# TEXT2FACE: 3D MORPHABLE FACES FROM TEXT

**Will Rowan**[1]**, Patrik Huber**[1]**, Nick Pears**[1]**, Andrew Keeling**[2]
[1]University of York, [2]University of Leeds

## ABSTRACT

We present the first 3D morphable modelling approach, whereby 3D face shape can be directly and completely defined using a textual prompt. Building on work in multi-modal learning, we extend the FLAME head model to a common image-and-text latent space. This allows for direct 3D Morphable Model (3DMM) parameter generation and therefore shape manipulation from textual descriptions. Our method, Text2Face, has many applications; for example: generating police photofits where the input is already in natural language. It further enables multi-modal 3DMM image fitting to sketches and sculptures, as well as images.

## 1 INTRODUCTION

Generative 3D shape models, such as 3D Morphable Models (3DMMs) (Blanz & Vetter, 1999), are useful statistical priors with which to explain 2D/3D images of 3D objects, such as human faces, by reconstructing their shape. This has enabled applications in personalised avatar design (Lombardi et al., 2018), medical diagnosis of fetal alcohol syndrome (Suttie et al., 2013), and prosthesis design for missing facial regions (Mueller et al., 2011). Furthermore, there are a multitude of generative applications. For example, Blanz et al. (2006) generate police photofits from eyewitness accounts, but they rely on manual manipulation of 3DMM parameters. Instead, we propose a method which allows descriptive text to be directly mapped to the latent space of the 3DMM. In doing so, we enable photofits to be initialised directly from witnesses' textual descriptions or a sketch.

CLIP (Contrastive Language-Image Pre-training) (Radford et al., 2021) is a pre-trained visual-textual embedding model. This approach has shown strong zero-shot capabilities on a wide range of computer vision tasks and has enabled models to take advantage of this embedding space for downstream tasks, such as text-to-image generation (Ramesh et al., 2021). Several approaches have used the expressive power of CLIP to relate text to 3D shape. This includes text-driven generation of stylised meshes (Michel et al., 2022); general text to shape generation (Sanghi et al., 2022); and full body 3D avatar creation and animation (Hong et al., 2022). ClipFace Aneja et al. (2022) allows for text-based texture and expression editing of pre-existing parameterised faces. However, it necessitates a parameterised face as input and maintains a fixed identity. None of these methods consider the generation of a fully parameterised model of the human face, including identity, from text alone.

In this work, we bring together CLIP and 3DMMs to enable direct text to 3DMM generation. To do this, we train a deep MLP, Text2Face, to map from CLIP embedding space to the space of 3DMMs, and show the strong qualitative results of this approach.

## 2 PROPOSED METHOD

Our method enables us to generate a fully parameterised 3D model of the human head, including identity, expression, and a detail map, from a single text prompt. To do this, we first generate a dataset of mappings between CLIP embedding space and the parameter space of the FLAME model.

We synthesise $50,000$ adult faces using StyleGan2 (Karras et al., 2020), selecting images estimated to be older than 18 using py-agender (Butlitsky, 2018). A CLIP embedding is extracted from each image using the ViT-L/14-336px vision transformer model (Radford et al., 2021). We further estimate identity, pose, and expression vectors for each image in FLAME model space (Li et al., 2017)

---

{wjr508, patrik.huber, nick.pears}@york.ac.uk, a.j.keeling@leeds.ac.uk

using DECA (Feng et al., 2021), a state of the art method for monocular 3D face reconstruction. We extract identity, expression, pose, and a detailed displacement map, $\delta$, for each image.

We train a deep MLP, Text2Face, on this dataset to map the CLIP embedding space to the FLAME parameter space. At inference time, we take advantage of CLIP's interchangeable text-image latent space by using CLIP's text encoder on a sequence of input text. We supply this CLIP embedding as input to Text2Face which generates parameters for a fully parameterised 3D face. These are passed to the DECA decoder (Feng et al., 2021) to produce the 3D mesh. Hence, training Text2Face on embeddings extracted solely from images enables inference for both text and images. We further use DALL-E (Ramesh et al., 2021) to generate an image for each text prompt, using texture mapping to map this to the mesh. The overall architecture is shown in Figure 1.

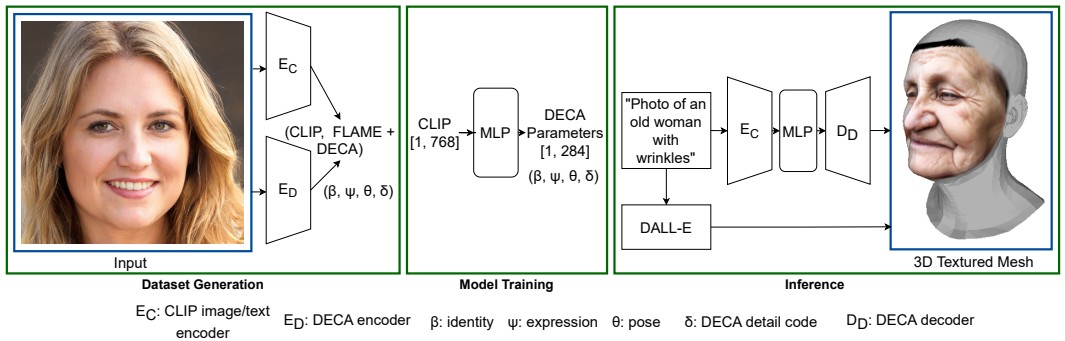

Figure 1: Dataset generation, model training, and inference for Text2Face.

We use the Adam optimiser (Kingma & Ba, 2014) with a learning rate of 1e-3 and a batch size of 64. We train for 100 epochs, using early stopping with a patience of 10. A full network diagram is presented in Appendix B (Figure 11).

## 3 EXPERIMENTS

Figure 1 shows the resulting textured 3D mesh from the text prompt: "Photo of an old woman with wrinkles". Figure 2 shows detailed texture-less meshes generated by the specified text prompts. The image generated by DALL-E from this same prompt, along with the subsequent textured mesh, are also shown. The pose is estimated from the generated image, and then applied to both meshes for visual comparison. Further qualitative results are shown in Appendix A.

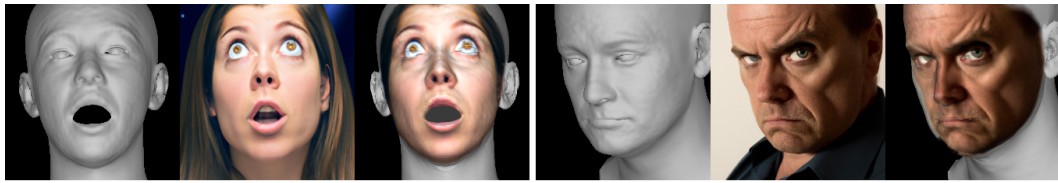

Figure 2: *(l)*: Prompt: "20 year old woman looking at the sky with surprise at UFO overhead". *(r)*: Prompt: "50 year old man looking grumpy". Each sub-figure, from left to right: Shape generated by the text prompt, DALL-E image from the same prompt, textured mesh.

## 4 CONCLUSION

We have presented the first method for text to fully parameterised 3D face shape. This finds application in photofit specification, avatar creation, and wider 3DMM fitting settings. Further work should consider inherited gender and racial biases from CLIP Agarwal et al. (2021), their impact on 3D face generation, and how this can be minimised.

## URM STATEMENT

The authors acknowledge that at least one key author of this work meets the URM criteria of ICLR 2023 Tiny Papers Track.

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

## A    FURTHER QUALITATIVE EVALUATION

In the following figures 3 to 9, we show three images. The first image shows the 3D mesh constructed from the 3DMM parameters regressed by Text2Face from the specified text prompt. The second image is an image generated by Ramesh et al. (2021) from the same text prompt. The final image shows this generated image mapped onto the generated mesh as texture. The pose for the meshes displayed here are estimated from the image generated by DALL-E; this allows for the texture to be fully displayed as shown. This full pipeline is implemented to enable a textured 3D mesh to be generated directly from a single text prompt.

The identity, expression, and detail code are all regressed directly by Text2Face from the text prompt.

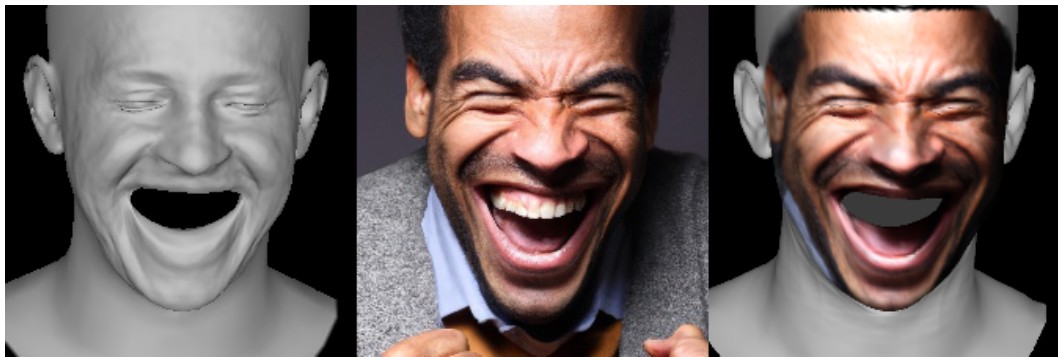

Figure 3: Prompt: "Happy man".

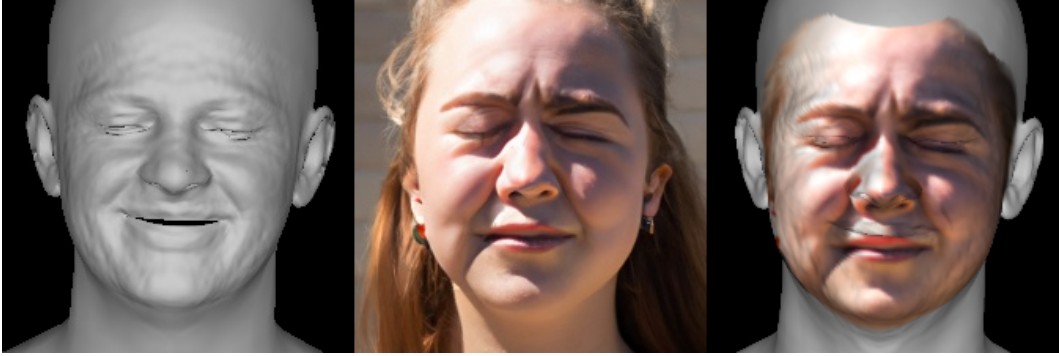

Figure 4: Prompt: "20 year old woman squinting at the sun".

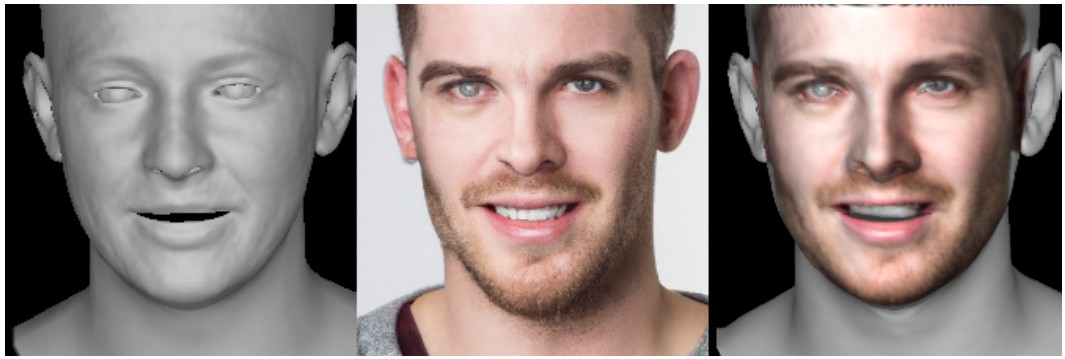

Figure 5: Prompt: "24 year old attractive man".

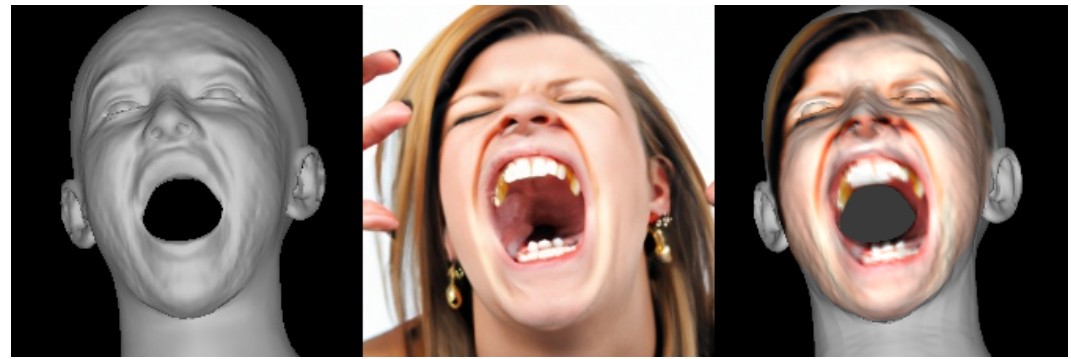

Figure 6: Prompt: "Photo of a woman screaming".

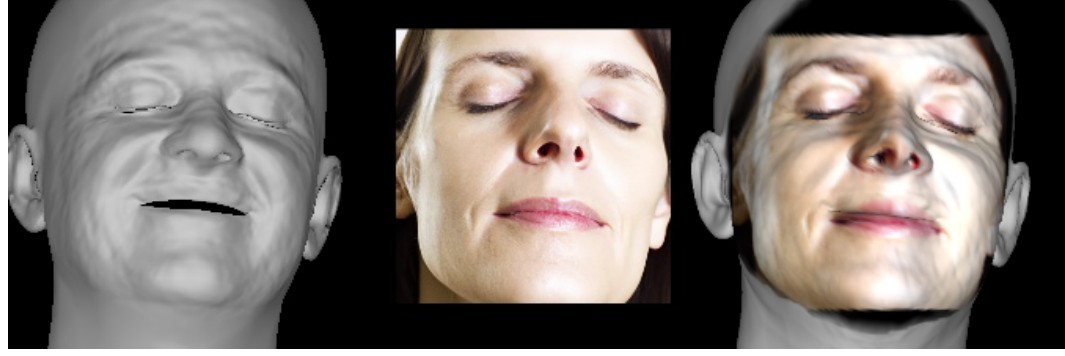

Figure 7: Prompt: "Photo of a woman with her eyes closed".

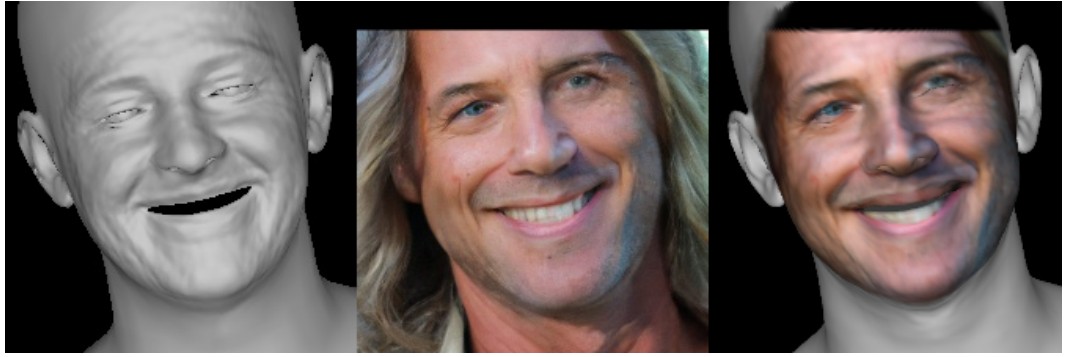

Figure 8: Prompt: "50 year old man looking happy after a long day working on the film set".

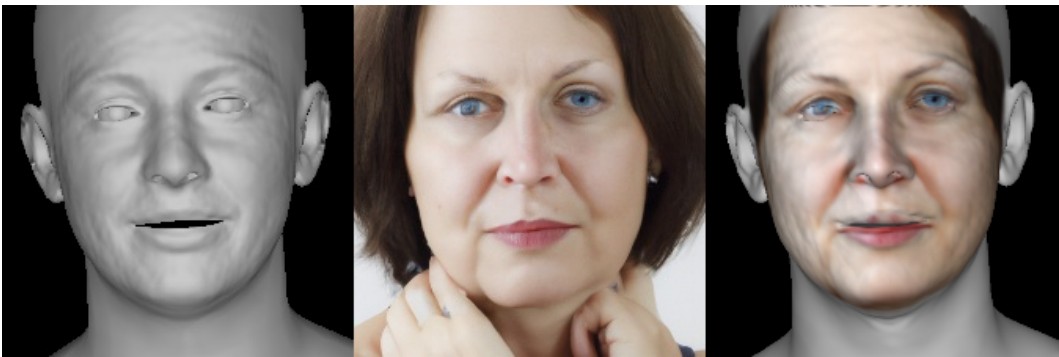

Figure 9: Prompt: "50 year old woman".

## A.1 MULTI-MODAL FITTING

Here we further demonstrate the ability of Text2Face to enable multi-modal input fitting to a 3DMM. To do this, we consider three related images. We take an image of Robert De Niro and create sketch and sculpture versions using image processing techniques. We extract the CLIP embedding from each image using the ViT-L/14-336px vision transformer model Radford et al. (2021) and pass this as input to Text2Face which regresses the 3DMM parameters, including identity, expression, and a personal detail code.

The result is shown in Figure 10. We estimate face pose from the original image of De Niro, giving all meshes this same pose to enable direct comparison. We also show texture mapping results for all generated meshes, using the first image to generate this texture.

Figure 10: Robert De Niro fit to a 3DMM using Text2Face: using an original image (left), a sketch (middle), and an engraving (right).

## B  TEXT2FACE ARCHITECTURE

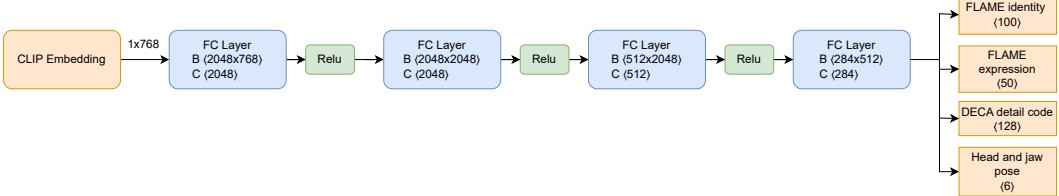

Figure 11: Architectural diagram of our Text2Face regressor.

