# OpenReview forum: "Text2Face: 3D Morphable Faces From Text"
_ICLR.cc/2023/TinyPapers — Submitted to Tiny Papers @ ICLR 2023_

### Official Review · Reviewer_UeCv · 2023-03-27

**Confidence:** 4

**Summary Of Contributions:**

The paper proposes a method to generate 3D deformable faces equipped with texture, starting solely from a text description. The method is based on a combination of CLIP, FLAME, DECA, and DALL-E. Several qualitative results are reported.

**Rating:**

Great Start (GS): a submission which meets some of the reviewing criteria but has room for improvement

**Strengths And Weaknesses:**

STRENGTHS
======
1) Unifying text and 3D Morphable Models is a hot topic at the moment, and such approaches have a tremendous potential impact in the future.
2) The reported qualitative examples provide some nice stress with both for the poses and identities of the considered subjects. This indeed suggests method flexibility

WEAKNESSES
========
1) The paper provides a nice discussion of previous work, but it would benefit from a better positioning, especially w.r.t. Aneja et al. 2022. This latter architecture highly resembles the one proposed in this paper, and I have a hard time understanding the main contribution from that. The paper suggests that "none of these methods considers the generation of a fully parametrised model of the human face, including identity", but Aneja et al. 2022 are significantly close. The main difference that I see is that their input texts seem quite limited, but given that they also rely on DECA, FLAME and CLIP I wonder what is actually the real methodological difference.
2) I see the paper obtains textured 3DMM, but no experiments are performed to show the deformation capability of the obtained model. This is a critical and important point to address because the results look quite interesting, and they may be even more exciting if some manipulations are provided.
3) Evaluating the generation capability of a network quantitatively is always challenging, but I think some numerical results are needed to validate the method's performance. Following the protocol of Aneja et al., I think that FID and KID might be good metrics which would provide convincing evaluations to the reader.

**Suggested Changes:**

To address the above weaknesses, I suggest the following changes:
1) Incorporate a discussion of Aneja et al. 2022, especially highlighting novelty and contribution w.r.t. to them. If it is possible, provide comparisons. Given that now the code is not available yet, it is also possible to test some of their qualitative prompts and compare the quality with the one provided by this method.
2) Add manipulation experiments, showing the capability of the obtained 3DMM and the consistency across different shapes and poses. This would strengthen the potential applications of the paper and point the actual advantage of learning a 3DMM.
3) Add quantitative analysis, with comparison w.r.t. a baseline. The most naive baseline could be a nearest-neighbour one (e.g., given a new text, we pick the nearest texture and mesh from the training dataset to show the generalization of the method), but of course, having something more sophisticated would be better.
4) I would also suggest having a more focused title since "multi-modal" sounds generic. To give a suggestions: "Text2Face: 3D Textured Morphable Faces from Text Input"

---

### Official Review · Reviewer_5ex5 · 2023-03-27

**Confidence:** 4

**Summary Of Contributions:**

This paper connects the space of CLIP image-caption embeddings to the parameter space of a 3DMM (FLAME + DECA), by training a deep MLP on a dataset of StyleGAN-2 generated faces and inferred (CLIP, FLAME+DECA) embedding pairs. This allows a 3D face to be initialized from a text description of the face.

**Rating:**

High Potential (HP): a submission which meets the reviewing criteria and has potential to make an impact on the field

**Strengths And Weaknesses:**

Strengths
* the paper is clearly written, well referenced, with nice figures
* the motivation for the paper is clear and the approach brings together and makes good use of a lot of recent models - CLIP, FLAME, DECA, StyleGAN2, DALL-E
* although not directly reproducible, the work is clearly explained enough that one would be able to set up a similar experiment
* the qualitative results presented show good promise across a variety of text prompts

Weaknesses
* the claim that "the method shows a better (than average) shape initialisation" should ideally be backed up quantitatively
* the detail about how "using texture mapping to map this to the mesh" could be provided -- it's a bit unclear in the paper and otherwise rather surprising that the DALL-E generated images have a similar pose


**Suggested Changes:**

Besides the clarification above and the few minor points below, I think this is an interesting paper which people will enjoy reading and thinking about in the workshop.

Minor
* Fig 1 - order the legend by order of appearance from left to right to make easier to scan. Or color-code things.
* Use `` for opening quotation marks
* I think it would be nice to show more examples of just the 3D generation (since this is the key contribution of the paper) in the main text, e.g. a grid in which certain specific attributes are varied. This might help to assess the ability of the model to support the generation of police photofits - as in that application, the witness might often say e.g. "the chin is more pointy", or "the cheekbones are more prominent" - i.e. exploring smaller variations of the 3DMM space with targeted textual feedback. This can tie the contribution of the model back to one of the original motivations, and show whether there is still more work to be done (which can be a good thing - it means there are more things to be discovered!), or the model might be already very useful.

---

### Official Review · Reviewer_q2J1 · 2023-03-28

**Confidence:** 4

**Summary Of Contributions:**

The paper presents the first 3D morphable modelling approach, where a 3D face shape can be defined using a textual prompt. This work extends the FLAME model to a common image-text space. This allows for direct shape manipulation from textual descriptions. Such work has many generative applications.

**Rating:**

High Potential (HP): a submission which meets the reviewing criteria and has potential to make an impact on the field

**Strengths And Weaknesses:**

Strengths
- Clearly defined research problem, motivation and background.
- The approach has been clearly explained.
- Discussion of future work on responsible AI considerations around face generation are absolutely crucial and it is great that the authors have recognized this.

Weakness
- The background could include more relevant citations. For example, [1] for text-to-image generation

[1] Yu, Jiahui, et al. "Scaling Autoregressive Models for Content-Rich Text-to-Image Generation." Transactions on Machine Learning Research.

**Suggested Changes:**

Add relevant prior work to the discussion.

---

### Author Response · Authors · 2023-05-31
**Response to Reviewers**

We thank the reviewers and area chair for their thoughtful and instructive feedback, and we are encouraged by their positive response to our paper.

Subsequently, we have positioned our work more distinctly in relation to Aneja et al. 2022 as suggested by reviewer UeCv, and we have refined the paper's title to be more precise. We have improved Figure 1 as recommended by reviewer 5ex5. Additionally, we now describe how we estimate pose for our visual comparisons, both within the main paper and in the appendix.

Both reviewers UeCv and 5ex5 have suggested further manipulation experiments to explore the model's response to targeted textual prompts. Given the space limitations of the TinyPaper format, we hope to prepare a more comprehensive submission that addresses these questions in the future.

Thank you again for your time and dedication to maintaining the high standard of research presented at the ICLR conference.

---

### Comment · Area_Chair_svSn · 2023-06-06
**Archival**

This work meets the threshold for archival, contains the URM statement (without specifying the author who meets the requirements) and is deanonymized.

---

### Meta-Review · Area_Chair_svSn · 2023-04-03

**Recommendation:** Invite to archive
**Confidence:** 4

**Metareview:**

I see the following points from the reviews:
- Clarity: reviewers agree that the paper is overall clear, while some further discussion w.r.t. related works is needed
- Correctness: reviewers agree that the paper sounds overall correct, while some further experiments would be required to back up the paper's claims, in particular from a quantitative perspective
- Reproducibility: the paper combines several different advanced tools, which might be difficult to replicate exactly without the code, but in general, the details seem enough to set up a similar experimental setting.
- Basic requirements: the paper meets the basic requirements of the submission.

The problem is generally considered attractive, and the method has a high potential; however, some missing experiments and positioning in the related work leave concerns about the clarity and correctness of the paper.

**Summary:**

The paper presents a method to generate textured 3D morphable models from textual prompts. The paper addresses an interesting problem and has a recognized high-potential, but some related work are not properly discussed and quantitative experiments are missing.

**Comments And Feedback To The Authors:**

I want to encourage the authors to follow this research direction, pointing out that the work has collected excitement among the reviewers. In the current state, some crucial missing make it challenging to assess the clarity and correctness of the paper firmly. Providing more quantitative evidence and taking similar works as inspiration would be a good way to understand the actual capabilities better. Discussing related work (with particular attention to highlighting the contribution w.r.t. concurrent works, grounding it with proper experiments) and fixing the minor presentation issues pointed out by Reviewer 5ex5 would also be significantly beneficial. Reconsidering the title is a good option to shape better the paper's focus. I do not doubt that addressing these points in the paper will be of great potential impact.

**Reason For Not Giving A Higher Recommendation:**

While the qualitative results look promising, the missing quantitative analysis seems to be a significant drawback which limits the possibility of assessing the applicability and correctness of the method. Also, given the large attention collected by text-to-3D methods, it is crucial to precisely place the method among other alternatives, well-describing differences.

**Reason For Not Giving A Lower Recommendation:**

The method has collected interest among the reviewers and addresses a cutting-edge problem. The qualitative results show a good variety, which seems compelling. The paper meets most of the requirements of clarity, reproducibility, and correctness.

---

### Decision · Program_Chairs · 2023-04-10

Invite to archive